# How Telemedicine Can Improve the Quality of Care for Patients with Alzheimer’s Disease and Related Dementias? A Narrative Review

**DOI:** 10.3390/medicina58121705

**Published:** 2022-11-22

**Authors:** Efthalia Angelopoulou, Nikolaos Papachristou, Anastasia Bougea, Evangelia Stanitsa, Dionysia Kontaxopoulou, Stella Fragkiadaki, Dimosthenis Pavlou, Christos Koros, Yıldız Değirmenci, John Papatriantafyllou, Eleftherios Thireos, Antonios Politis, Agis Tsouros, Panagiotis Bamidis, Leonidas Stefanis, Sokratis Papageorgiou

**Affiliations:** 1Department of Neurology, Aiginition University Hospital, National and Kapodistrian University of Athens, 11528 Athens, Greece; 2Medical Physics and Digital Innovation Lab, Aristotle University of Thessaloniki, 54636 Thessaloniki, Greece; 3School of Topography and Geoinformatics, University of West Attica, 12243 Aigalew, Greece; 4Neurology Department, Istanbul Medipol University Faculty of Medicine, 34214 Istanbul, Turkey; 5Primary Health Center of Vari, National Health System of Greece, 16672 Athens, Greece; 6Department of Global Health, Boston University School of Public Health, Boston, MA 02118, USA

**Keywords:** Alzheimer’s disease (AD), telemedicine, COVID-19, World Health Organization (WHO)

## Abstract

*Background and Objectives*: Dementia affects more than 55 million patients worldwide, with a significant societal, economic, and psychological impact. However, many patients with Alzheimer’s disease (AD) and other related dementias have limited access to effective and individualized treatment. Care provision for dementia is often unequal, fragmented, and inefficient. The COVID-19 pandemic accelerated telemedicine use, which holds promising potential for addressing this important gap. In this narrative review, we aim to analyze and discuss how telemedicine can improve the quality of healthcare for AD and related dementias in a structured manner, based on the seven dimensions of healthcare quality defined by the World Health Organization (WHO), 2018: effectiveness, safety, people-centeredness, timeliness, equitability, integrated care, and efficiency. *Materials and Methods*: MEDLINE and Scopus databases were searched for peer-reviewed articles investigating the role of telemedicine in the quality of care for patients with dementia. A narrative synthesis was based on the seven WHO dimensions. *Results*: Most studies indicate that telemedicine is a valuable tool for AD and related dementias: it can improve effectiveness (better access to specialized care, accurate diagnosis, evidence-based treatment, avoidance of preventable hospitalizations), timeliness (reduction of waiting times and unnecessary transportation), patient-centeredness (personalized care for needs and values), safety (appropriate treatment, reduction of infection risk),integrated care (interdisciplinary approach through several dementia-related services), efficiency (mainly cost-effectiveness) and equitability (overcoming geographical barriers, cultural diversities). However, digital illiteracy, legal and organizational issues, as well as limited awareness, are significant potential barriers. *Conclusions*: Telemedicine may significantly improve all aspects of the quality of care for patients with dementia. However, future longitudinal studies with control groups including participants of a wide educational level spectrum will aid in our deeper understanding of the real impact of telemedicine in quality care for this population.

## 1. Introduction

Dementia is a clinical syndrome characterized by cognitive decline leading to impaired functional activities of daily life [1]. Alzheimer’s disease (AD) is the most common cause of dementia, followed by Lewy body dementia and Parkinson’s disease, frontotemporal dementia, vascular dementia, and other rarer underlying conditions [1]. It is estimated that dementia affects more than 55 million patients worldwide, with a significant societal, economic, and psychological impact not only on the patients themselves but also on their family members, caregivers, and health system [2]. Due to the increased life expectancy and aging population, the prevalence of dementia is expected to grow continuously. 

Unfortunately, there is still no universally approved disease-modifying efficient treatment for AD and other neurodegenerative causes of dementia [2]. However, great efforts have been made toward the development of effective ways of care for patients with cognitive impairment. Appropriate symptomatic treatment, management of behavioral symptoms, support and education of caregivers, engagement in social activities, counseling, home modifications, and the use of non-pharmacological treatments have been shown to improve quality of life [3]. 

Even though early diagnosis has been associated with improved quality of life and treatment, literature evidence shows that many patients with AD and other forms of dementia have not received a formal diagnosis yet [4]. Diagnosis is often even more delayed in patients with early disease onset (younger than 65 years of age) [5]. Despite the availability of resources and services, many patients have inadequate access to appropriate treatment, specialized care, and holistic support [4]. In comparison to other age-related disorders, the quality of healthcare for patients with AD and related dementias is poor [6]. Potential underlying reasons include time constraints in medical practice, the lack of specialized neurologists, psychiatrists, and geriatricians in rural areas, the lack of experience and education of primary care physicians, the fragmented health care services, stigma, and the limited integration of community resources in dementia care [6,7]. Furthermore, intrinsic factors related to dementia, including limited recognition of medical needs, difficulties in communicating health problems and navigating health systems, may also lead to additional challenges in reaching appropriate care [8]. High-quality healthcare is a fundamental goal of the healthcare system. However, the observed inequity in the receipt of care indicates that current healthcare services are rather inadequate to holistically and efficiently address the needs of the patients. For these reasons, the World Health Organization (WHO) recognizes dementia as a public health priority, and alternative approaches to dementia care are urgently needed [9].

Telemedicine, defined as the remote diagnosis and treatment of patients via information and communications technology, holds promising potential for addressing this important gap [10,11]. Even though telemedicine has been already used for decades, the COVID-19 pandemic highlighted the emerging need for remote care, especially for the vulnerable population with chronic diseases, including patients with cognitive impairment. Physical distancing for preventing infection risk, increased caregivers’ burden, overload of the healthcare system, and suspension of medical visits for non-urgent chronic conditions have contributed to the acceleration of remote care.

The reliability of telemedicine in cognitive impairment for diagnosis and follow-up, as well as the facilitators and barriers of this type of service in dementia care, have already been discussed elsewhere [7,12,13,14]. However, there is no literature review analyzing the ways in which telemedicine could improve the quality of care for patients with dementia in a structured manner. The increasingly growing research in this field and the rapid adoption of telemedicine in clinical practice necessitate a comprehensive analysis of the role of telemedicine in quality of care based on a commonly used conceptual framework, as well as a critical consideration of potential related challenges.

Herein, we analyze recent evidence on how telemedicine can improve the quality of care for patients with AD and related dementias. For this purpose, our discussion is based on the seven dimensions of healthcare quality defined by WHO (2018): effectiveness, safety, people-centeredness, timeliness, equitability, integrated care, and efficiency [15]. Next, we discuss the potential challenges and opportunities of telemedicine, especially for patients with dementia, aiming for a better understanding of its role in this specific disease population. Finally, we provide perspectives for its effective implementation in the future.

## 2. Materials and Methods

Although our aim was not to conduct a systematic review, we followed the basic principles of a systematic review but limited to published peer-reviewed academic literature and a narrative synthesis of findings, as previously described [16,17,18]. We searched MEDLINE and Scopus databases for peer-reviewed articles investigating the role of telemedicine in the quality of care for patients with dementia written in the English language, with no time restrictions. The search was conducted between March 2022 and August 2022. We used the terms “dementia”, “Alzheimer’s disease”, “cognitive impairment”, “cognitive decline”, “memory impairment”, “healthcare”, “care”, “quality of care”, “telemedicine”, “telecare”, “teleneurology”, “tele-neurology”, “remote care”, “telehealth”, “access”, “accessibility”, “effective”, “effectiveness”, “patient-centered”, “personalized”, “individualized”, “integrated”, “safe”, “safety”, “equal”, “equitability”, “equity”, “timely”, “timeliness”, “efficient”, and “efficiency” in different combinations. Relevant articles were screened in the title and abstract, and relevant articles were read in their full form. We included articles mentioning the role of telemedicine in the quality of care of patients with AD or related forms of dementia in terms of effectiveness, safety, equitability, timeliness, patient-centeredness, efficiency, and integrated care. Studies without mentioning results, studies among patients under the age of 18 years old, or those investigating intellectual disability were excluded. Through the snowballing process, we also screened the bibliography of each selected article for potential additional studies to include most of the key recent evidence [19]. For the purpose of this review, we organized our narrative synthesis of the included studies by the thematic categories defined by WHO on healthcare quality (1) effectiveness, (2) safety, (3) patient-centeredness, (4) timeliness, (5) equitability, (6) integrated care, and (7) efficiency.

## 3. Results

### 3.1. Defining the Concept and Dimensions of Healthcare Quality

Until now, there has been no widely accepted definition of “quality of healthcare”; however, there is a commonly shared comprehension of the main aspects and aims of healthcare quality, which includes the delivery of effective and safe care for the improvement of patients’ welfare [15].

In order to explore how a new healthcare delivery model could improve the quality of care, we should first understand the dimensions of healthcare quality in an organized and structured manner. According to the conceptual framework of the United States Institute of Medicine (IOM), the quality of healthcare is defined as “the degree to which health services for individuals and populations increase the likelihood of desired health outcomes and are consistent with current professional knowledge” [20]. This definition is widely adopted among healthcare stakeholders and highlights the importance of contemporary evidence-based care provision for positive outcomes at an individual and population level. Furthermore, IOM describes the six main components of healthcare quality: effectiveness, safety, patient-centeredness, timeliness, efficiency, and equitability [20]. As mentioned in the IOM report, this useful framework enables us to conceptualize better the main dimensions of healthcare quality [20], and it is currently used in many countries other than the United States [15]. 

Similarly, as recently stated in the World Health Organization (WHO) Framework on Integrated People-centered Health Services—which was based on the IOM framework—“high-quality care” involves “care that is safe, effective, people-centered, timely, efficient, equitable and integrated” (Figure 1) [15].

As explained by WHO, the effectiveness of care includes the provision of evidence-based health services to individuals who need them, and safety is defined by the avoidance of harm. People-centeredness means the delivery of care that respects and responds to personal needs, values, and preferences [15]. Timely healthcare provision implies the reduction of waiting times and delays that could be harmful not only for those who receive but also give care [15]. Equitable health services are those whose care quality is independent of age, race, ethnicity, geographical location, socioeconomic status, sex, gender, religion, and political or linguistic affiliation [15]. Integrated care is described as coordinated care across different levels and providers, allowing the availability of all appropriate health services throughout the life course [15]. Finally, efficiency in healthcare is defined as maximizing the benefit of available care resources and avoiding relevant waste [15]. 

In the following sections, we explore the potential of telemedicine to improve healthcare quality for patients with AD and other forms of dementia based on these seven dimensions of healthcare quality as described by WHO and further discuss future opportunities. 

### 3.2. Definition and Primary Forms of Telemedicine 

As mentioned above, telemedicine is defined as the remote diagnosis and treatment of patients using information and communication technology [10,11]. Even though the terms “telehealth” and “telemedicine” are used interchangeably in many cases, “telehealth” is a broader term incorporating additional remote non-clinical services, such as clinician training, medical education, and administrative meetings [21]. 

Essentially, there are three main types of telemedicine services: synchronous, asynchronous, and remote monitoring. Synchronous telemedicine involves the delivery of care in real-time, allowing for live interaction with the patient or the physician to provide expertise. A subtype of synchronous telemedicine visit involves the Facilitated Virtual Visit. An example of this type of visit is when the patient is at a site where diagnostic equipment is available (i.e., pulse oximeter, digital stethoscope) and the physician is at a remote site. At the patient’s site, the telefacilitator (i.e., nurse) gathers objective medical measurements and transfers these data to the remote physician. Asynchronous telemedicine involves the “store-and-forward” technique, such as the transfer of prerecorded neuroimaging data for review by neuroradiologists [22]. Remote patient monitoring refers to the continuous assessment of a patient’s clinical condition via direct video monitoring or review of various tests and images being collected remotely. 

Telemedicine services are provided through a wide variety of applications, including telephone, video-conferencing, communication via e-mail, mobile applications, and the use of remote devices, such as wearable biosensors [23]. Wearable biosensors are multiplexed, smart devices that enable the non-invasive quantification of several dynamic biological signals in real time via optical, mechanical, and electrochemical modes of transduction. These approaches allow for integrated and multifaceted data acquisition and interpretation for personalized healthcare monitoring.

Teleconsultation refers to the cases when healthcare providers present a patient’s case to medical experts in another remote location—usually at a hospital or specialized clinic—asking for expert consultation. In this case, the patients may or may not be present during the video conferencing. In other cases, telemedicine is provided by healthcare professionals directly to patients [21].

### 3.3. The Reliability of Telemedicine in AD and Related Dementias

Given that the evaluation of patients with AD and other types of dementia is majorly based on the medical history and clinical interviews with the patient and family members, telemedicine is a precious tool for these cases [7]. The diagnostic accuracy of dementia is comparable to traditional in-person examination [13]. Although several parts of the neurological examination, such as gait assessment, can be conducted remotely, other aspects of the exam, such as tone, deep tendon reflexes, and sensory and muscle strength examination, may be challenging to evaluate without a healthcare professional close to the patient. Neuropsychological testing is generally feasible and reliable through telemedicine. In particular, Mini-Mental State Examination (MMSE) can be reliably administered in patients with cognitive impairment [24]. Montreal Cognitive Assessment (MoCA) and a modified version for video-based conditions have been proven reliable if administered remotely in individuals with cognitive complaints or mild-to-severe AD, respectively [25,26]. Although no significant differences have been detected between in-person and video-based performance on MMSE and Alzheimer’s Disease Assessment Scale cognitive subscale (ADAS-cog) of patients with AD during two-year period, individuals at advanced stages performed worse in some cases of video-based assessment [24]. The administration of other cognitive scales such as Repeatable Battery for the Assessment of Neuropsychological Status (RBANS) [27], language testing with Boston Naming Test and Letter and Category Fluency, memory testing with Hopkins Verbal Learning Test-Revised, attention and working memory testing with Digit Span forward and backward, as well as Clock Drawing Test and Visuospatial Memory Test can also be administrated reliably through telemedicine [12,28,29,30]. The assessment of AD staging with the Clinical Dementia Rating scale (CDR) [26] and depressive symptoms with the Geriatric Depression Scale (GDS) is also reliable [31]. Telephone-based instruments, such as Telephone Interview for Cognitive Status (TICS), are also helpful in assessing cognitive function remotely [32]. In addition, the use of automatic speech analysis for the diagnosis of dementia or other related digital tools may also help for the remote assessment of cognitive complaints [33]. 

### 3.4. Quality of Care for AD and Related Dementias and the Emerging Role of Telemedicine: Current Evidence

In this section, we summarize the importance of healthcare quality for patients with AD and other forms of dementia in each of the seven dimensions as defined by WHO (effectiveness, safety, people-centeredness, timeliness, equitability, integrated care, and efficiency, Table 1). We further discuss the emerging role of telemedicine in addressing the existing gaps (Table 2).

#### 3.4.1. Effectiveness

Although the actual frequency of delayed or missed diagnosis of AD and related dementias remains unknown, it is estimated to be particularly high in primary care settings [4]. Many primary care physicians are not familiarized with the diagnostic criteria of dementia, and they lack appropriate education or expertise in assessing and treating patients with cognitive impairment [4,34]. Neurodegenerative causes of dementia should be differentiated from potentially reversible ones, including depression, infections, metabolic disorders, central nervous system tumors, autoimmune conditions, and functional cognitive disorders [35]. Dementia misdiagnosis may result in significant harm, so the correct discrimination of dementia mimics is very important [36]. Recognizing atypical clinical presentations and rarer dementia forms, especially at younger ages, such as frontotemporal dementia, is even more challenging [36]. The most commonly used cognitive tests lack sensitivity, and diagnosis may be missed for patients at early stages. Hence, detailed neuropsychological testing is often required [37].

Furthermore, the evaluation of patients with cognitive complaints requires significant time. Within the often-busy primary care settings, the physicians may not be able to sufficiently discuss with the family, provide counseling to caregivers, give referrals to the appropriate community-based organizations, and develop a personalized management plan [34]. Adherence to dementia guidelines and evidence-based recommendations has been associated with a better overall quality of care, patient health-related quality of life, and quality of caregiving [38]. However, several studies have demonstrated that the adherence of physicians to dementia guidelines and quality care indicators is inadequate in primary care settings concerning assessment, treatment, support, education, and safety [38,39,40]. As a result, most patients with dementia and their caregivers find it difficult to access formal care services, and when reached, this care is not the appropriate one [41]. In addition, the participation of physicians in education seminars about common issues in dementia care has been associated with improved quality of life of the patients [38].

In this context, telemedicine provides the opportunity for improved access to appropriate and specialized care. Telemedicine has been shown to be effective in confirming or providing a diagnosis in case of cognitive impairment [62,74,76,137,138,139], highlighting its significance, especially for remote, underserved areas. Regional community clinics in rural areas can be effectively connected through teleconsultations with physicians who are specialized in dementia care in University Hospitals. In this way, accurate diagnosis is facilitated, and an appropriate treatment plan is provided [63]. Telephone-based remote care is feasible for younger patients with dementia, too [64]. Telemedicine could also reduce the risk of behavioral and psychological symptoms of dementia related to the negative consequences of the COVID-19 pandemic [140]. Telemedicine provided by specialists has been associated with 1.8 and 1.1 medication changes on average for patients with dementia at initial assessments and follow-up visits, respectively, during a 12-month period [65]. In another study, alterations in drug prescriptions were recommended in more than one-third of the patients through telemedicine [62]. Telemedicine consultation in dementia care has also been associated with treatment modifications at approximately 10% in another study, especially for those with AD or living with a relative [66]. Telemedicine use has been associated with longer treatment duration and compliance in dementia patients [67]; it has also been proven a feasible method for follow-up and ongoing care [50,68]. Via telemedicine, the cancellations of medical visits are fewer, and the transitions between the follow-up clinic and primary care are also improved [68]. These results highlight the vital role of telemedicine in providing appropriate care by prescribing the proper medications for each patient and improving treatment compliance. 

Through appropriate referrals by dementia specialists, telemedicine may also improve the patients’ access to diagnostic work-up, thereby aiding in the correct diagnosis. For example, after a telemedicine assessment by dementia specialists, a lumbar puncture may be recommended and conducted at the peripheral hospital. Then, the cerebrospinal fluid sample could be transferred for biomarker analysis to the appropriate laboratories, whose availability in remote areas is limited. In addition, images of Magnetic Resonance Imaging (MRI) can be transferred remotely to specialized neuroradiologists, allowing for a more accurate evaluation. Finally, after specialized assessment through telemedicine, genetic testing may be recommended, especially in early-onset, familial, or atypical cases, and blood samples could be transferred to the appropriate laboratories. These opportunities are especially important for residents living in remote areas with limited access to specialized care and guidance. 

Overall, telemedicine visits through video-conferencing for patients with neurocognitive disorders have been associated with improved quality of life, better physical and mental health, less perceived burden, and higher self-efficacy [42,69]. Telemedicine via video-conferencing may also improve the well-being and resilience of patients with neurocognitive disorders and caregivers [60]. Furthermore, the addition of telemedicine has been associated with a delayed deterioration of MoCA scores compared to only telephone-based visits [42]. Even though research evidence on dementia subtypes other than AD is limited, it has been demonstrated that telemedicine is a valid triage tool for patients with frontotemporal dementia [70], highlighting its promising potential for other forms of dementia. 

Importantly, dementia is associated with higher rates of hospital admissions attributed to ambulatory care-sensitive conditions, for which appropriate evaluation and early management in outpatient settings might have possibly prevented hospitalization [141]. Recent evidence shows that the number of plausibly avoidable hospital admissions of aged individuals with dementia is growing [142]. In rural areas, AD is associated with even higher preventable hospitalizations [143]. Usual underlying pathological conditions are pneumonia, congestive heart failure, and urinary tract infection, among others. Therefore, care for patients with dementia necessitates high-quality healthcare outpatient services for better outcomes and the prevention of avoidable visits to the emergency departments. 

In this context, telemedicine has been shown to be a valuable tool for the avoidance of unnecessary visits to emergency departments [71]. Emergency department use has also decreased for ambulatory-care sensitive conditions after the introduction of telemedicine for older individuals in senior living communities in another study [72]. Hence, telemedicine may also aid in the reduction of potentially unnecessary emergency visits and hospitalizations, allowing for cost-effective care not only for the patients and their families but also for the health systems.

Concerning education, video-conferenced geriatric medicine grand rounds on a weekly basis are feasible, acceptable, and beneficial for healthcare professionals, who otherwise could not have access to those medical rounds [73]. This opportunity allows for the tele-education of primary care physicians on dementia care, which could subsequently benefit patients. 

#### 3.4.2. Safety

Regarding safety concerns, the cognitive and mobility impairment of patients with dementia poses several challenges during their traveling to specialized physicians for in-person visits [7]. Therefore, telemedicine is a valuable tool since it can reduce the risk of accidents during transportation [144].

Inappropriate and high prescriptions of antipsychotic drugs for behavioral symptoms, as well as inadequate review and monitoring of medications for patients with dementia, are high [43,44]. Improper use of antipsychotics has been associated with a higher risk of death and ischemic events, especially in the elderly [45]. In this regard, a study has demonstrated that specialists could identify inappropriate drug use that could potentially contribute to cognitive decline in almost half of the visits through telemedicine [74]. This evidence highlights the significant role of telemedicine in detecting treatment approaches that could potentially harm patients with dementia. 

Furthermore, in the case of home-based video-conferencing, telemedicine gives physicians the opportunity to directly observe the home environment of the patient and suggest alterations that could improve daily life and safety. For instance, physicians could make individualized recommendations for the prevention of falls [75]. In addition, social determinants of health, such as family dynamics and economic difficulties, can also be more effectively detected in video-based visits at home [22], allowing for a more holistic approach to dementia care.

COVID-19 poses high infection risks in traditional in-person clinical settings and affects elderly individuals disproportionately [33]. For this reason, during the COVID-19 pandemic, patients are generally encouraged to use telemedicine services for safer medical assessment and management when possible. This is particularly important for the vulnerable elderly population with chronic diseases, which are related to higher COVID-19-associated morbidity and mortality [33]. During the COVID-19 pandemic, older patients with dementia are also at higher risk of not receiving appropriate medical care [42]. AD diagnosis has been independently associated with higher COVID-19-associated mortality [46]. Therefore, the safer environment of telemedicine that protects against COVID-19 transmission and reduces exposure risk provides a valuable option for dementia care, also allowing the continuity of care that these individuals need. The avoidance of COVID-19-related hospitalizations also prevents secondary infections and other complications that have also been associated with increased mortality [145]. During the COVID-19 pandemic, non-urgent outpatient visits for chronic diseases were suspended, leading to a sense of abandonment because of a lack of physician–patient contact [52]. In this context, telemedicine through video-conferencing was associated with better quality of life for patients with dementia compared to only telephone-based visits during the social isolation of the COVID-19 pandemic, highlighting its role in minimizing the potential adverse effects of social distancing measures [60].

#### 3.4.3. People-Centeredness

As demonstrated by the results of the UCLA Alzheimer’s and Dementia Care (ADC) Program, the individualized evaluation of the specific needs of patients and their families, as well as the adoption of a personalized management plan, are associated with higher-quality dementia care, regarding screening, assessment, and counseling [34]. Care models using shared decision making between patients, family members, and caregivers have also been associated with higher satisfaction from patients and caregivers [49]. In addition, individualized pharmacological and non-pharmacological interventions by primary care physicians in collaboration with dementia care managers have been associated with lower caregiver stress and behavioral symptoms, as well as possibly fewer inpatient hospitalizations [6,47,48].

Patients’ and family members’ preferences, perceptions, and needs are an important integral part of dementia care. In this regard, patients and caregivers generally accept telemedicine, and they perceive it as a very convenient model of care. Patients with AD and their caregivers are very satisfied with telemedicine, with overall satisfaction rates ranging between 88–98% [62,76,77,78]. Similar satisfaction rates have been observed between telemedicine and traditional in-person visits in one study [80]. However, telemedicine was preferred over in-person in other studies [76,79]. Importantly, in most studies, response rates on satisfaction surveys are generally low, suggesting that selection bias may lead to an overestimation of participants’ satisfaction [7].

Telemedicine allows the personalized assessment of individuals with cognitive impairment and may aid in developing and establishing an individualized treatment plan. Compared to primary care settings, telemedicine gives more time for discussion with patients, caregivers, and family members regarding their beliefs and needs [81]. A recent study investigating the experiences of patients with dementia and their caregivers in remote healthcare via semi-structured interviews demonstrated that proactive teleconsultations during the COVID-19 pandemic were effective. However, this study demonstrated that these teleconsultations should be more focused on real needs, practical recommendations, and ways to replace non-verbal prompts, especially for the description of new health problems [82].

Telemedicine also gives the opportunity to family members living away from the patient’s home to attend the video-conference visit, allowing for shared decision making a personalized treatment plan [83].

The use of smart home systems and remote monitoring devices allows older adults with cognitive impairment to live in their preferred environment, which may also delay their placement in nursing homes. In this way, patients’ and family members’ preferences are respected, and independent living with a sense of safety is facilitated [146]. On the other hand, another study indicated that the use of assistive technology and telecare for individuals with dementia was not associated with prolonged time of independent living [84]. Further studies are needed considering also dementia stage as an important factor that could influence the effects of assistive digital technologies. 

#### 3.4.4. Timeliness

Specialized neurologists and memory clinics are often unavailable in remote and rural areas [50]. On the contrary, AD prevalence has been shown to be higher in rural regions [51]. Telemedicine reduces waiting times for appointments with specialized physicians, thereby contributing to earlier diagnosis and timely treatment of dementia-related various medical complaints [60].

Earlier recognition of cognitive decline is associated with improved health outcomes [49]. A systematic review demonstrated that the diagnostic sensitivity of dementia is associated with the frequency of contact between patients and providers [4]. Regular monitoring contributes to a better quality of life [38,43], highlighting the importance of timeliness in a better quality of care. Via telemedicine, patients with dementia have the opportunity to receive regular follow-ups without the need to cancel scheduled visits for reasons related to travel restrictions. 

Telemedicine also gives the opportunity for real-time medical reporting and sharing, thereby avoiding unnecessary delays that could affect the quality of care [24]. Furthermore, sharing brain imaging data or laboratory results in an asynchronous manner accelerates the assessment process and facilitates the prompt recognition of other medical conditions [33]. The use of wearable devices, remote monitoring sensors, or web-based platforms may also be beneficial tools for the early detection of potential medical emergencies and timely intervention [33,85].

#### 3.4.5. Equitability

Equitability in quality and access are integral parts of healthcare delivery. Health inequality is defined as “differences in the distribution of health status and achievement of health outcomes that exist among specific groups due to genetic or other factors that cannot be prevented or modified” [147]. Factors contributing to inequity are associated with differences in availability, cost, and access to information for various population groups [148]. The “European Dementia Monitor” Project indicated important differences in organizational, financial, and practical aspects across European countries regarding accessibility to dementia care and treatment, resulting in inequity [148]. Race, ethnicity, age, gender, educational level, and geographical area may contribute to health inequalities [147]. It has been demonstrated that the area of residence plays a vital role in accessibility [148]. Patients in rural areas often have to travel long distances to obtain appropriate access to specialized care [53]. The lack of transportation infrastructure and socioeconomic disparities may also limit the accessibility of patients in rural areas [52]. It is also difficult to recruit and retain physicians at rural health centers or hospitals [22]. Concerning race and ethnicity, compared to non-Hispanic Whites, a greater percentage of non-Hispanic Blacks and Hispanics had a delayed or missed diagnosis of dementia [54]. Potential reasons for this situation include disparities in health insurance coverage, different proximity to health services, racism, mistrust of the health system, and limited diversity in the healthcare personnel [54].

Telemedicine facilitates the elimination of geographical disparities since it gives equal opportunities for patients with dementia from rural and urban areas to access specialized healthcare [53,88], given the geographic misdistribution of medical specialties, including neurologists, telemedicine aids in eliminating this gap. Several studies have shown that telemedicine in rural areas is effective, with high satisfaction rates, allowing for better access to timely care, reduced cost, and avoidance of unnecessary transportation [22,53,149]. Telehealth also gives the opportunity to inhabitants of underserved and remote areas to be educated about health issues [22], including dementia prevention and care. Furthermore, telemedicine, offered by healthcare professionals trained in recognition of cultural diversities and needs of each person, can provide care to individuals living in underrepresented ethnic and racial communities, thereby contributing to the limitation of gaps in equitability.

On the other hand, inadequate experience with technology and digital literacy, lower education level, as well as a worse cognitive function have been associated with less engagement in remote interventions promoting lifestyle modifications among older adults [89,90]. Further attempts to train patients in the use of digital technologies are required to address healthcare inequalities related to the use of telemedicine interventions, especially among older individuals [150]. 

#### 3.4.6. Integrated Care

Apart from their purely medical needs, patients with dementia, caregivers, and family members often require psychological support, referral for legal issues, advice on the selection of the appropriate long-term facilities, and discussion about solutions to improve the home environment to facilitate their living and reduce the risk of falls. In addition, patients and family members need information about available social services, daycare centers, activities, support groups, and educational resources, as well as administrative assistance during the application for long-term services or nursing homes [34]. In addition, fragmented care and difficult-to-navigate healthcare services are significant barriers to effective treatment [57]. Hence, dementia care requires a holistic and multidisciplinary approach, which should involve the integration of several different stakeholders and organizations.

The integration of community-based organizations (e.g., Alzheimer’s Association) into the health systems has been shown to improve quality care for patients with dementia [55]. Transdisciplinary team care from physicians, neuropsychologists, social workers, registered nurses, and nurse practitioner managers has been associated with better quality of care, improved driving counseling, better caregivers’ counseling, reduced levels of caregivers’ stress, fewer patients’ behavioral and depressive symptoms, as well as fewer hospitalizations and visits to the emergency departments [38,55,56]. Linkages to appropriate community resources have been shown to be beneficial. The participation of nurse practitioners in care has also been associated with higher healthcare quality for dementia patients, reduced risk of falls and incontinence, as well as better adherence to care recommendations [58,59]. Social worker engagement with home assessments may also improve the quality of dementia care [38]. In integrated care models, team members contribute with their expertise and clinical or management strengths in a collaborative manner to provide the most appropriate dementia care approach [49]. However, in general, the cooperation between different healthcare professionals for the care of each patient with dementia is still limited, and community-based organizations are currently inadequately incorporated into the healthcare system [34,43].

In this regard, research has shown that telemedicine may significantly enable interdisciplinary dementia care [86,87,91,92,93]. In particular, a Pittsburgh-based telemedicine program for dementia care, including a geriatrician, geriatric psychiatrist, psychologist, social worker, and nurse manager, was highly acceptable and successful fur rural areas [74]. In a Tennessee-based program, specialists recommended referrals to social workers [76], as well as the use of long-term care services in almost two-thirds of the telemedicine visits [62]. This evidence suggests that telemedicine offers a significant opportunity for appropriate referrals to other services and useful consultation regarding decisions for long-term care. A study among patients with poor socioeconomic status and limited access to care has shown that the incorporation of a nurse practitioner-led mobile memory clinic into the general practice was feasible and acceptable [94]. Telemedicine can also aid in the assessment and management of psychotic symptoms of patients with neurodegenerative disorders in long-term care facilities [95]. In addition, specialists at different organizations and regions can easily connect to a telemedicine video conference and offer their expertise in a feasible and effective way, thereby contributing to the provision of integrated and holistic care.

Apart from direct physician–patient care, telemedicine also allows telerehabilitation for patients with AD dementia, frontotemporal dementia, and mild cognitive impairment [96,97,98,99]. Computerized cognitive training among patients with or at risk for dementia has been shown to be effective [100,101,102]. In particular, speech therapy has been proven effective in primary progressive aphasia [103] and alexia [104]. Virtual reality for patients with dementia has been associated with reduced neuropsychiatric symptoms such as depression and agitation [105], apathy [106], as well as improved quality of life [107]. Tele-exercise programs through video conferencing have been proven feasible and acceptable [108], as well as possibly effective in enhancing physical activity in patients with AD and their caregivers [109]. Therefore, telerehabilitation services may be integrated into dementia care and can be beneficial for patients in remote areas where these in-person facilities are lacking.

Furthermore, several studies have indicated that video-based caregiver support for stress, education, and training in managing patients’ behavioral symptoms is feasible and effective [110,111,112,113,114,115,116,117,118,119,120,121,122,123,124,125,126,127]. Remote cognitive behavioral therapy for the care givers of patients with AD for the enhancement of their physical or mental health [128] and for insomnia treatment [129] can also be effective. Most caregivers are satisfied with the FamTechCare service, which provides tailored expert feedback based on video recordings [130]. Telemedicine can also effectively educate caregivers about dementia management [52]. On the other hand, the subjective burden levels of the caregivers have not been significantly affected by a telehealth-based intervention in another study [131]. Furthermore, assistive technology and telecare were not associated with reduced caregivers’ burden [132]. Therefore, telemedicine allows for a more holistic approach to dementia care by integrating various services, healthcare professionals, and facilities in a feasible way. However, the partially contradictory results highlight the need for further studies that could aid in our deeper understanding of the long-term effects of remote care in patients with dementia and caregivers.

#### 3.4.7. Efficiency

Telemedicine has been shown to be very convenient for patients and their family members. Compared to traditional in-person visits, telemedicine can provide more flexibility regarding the time of the visit and limit potential alterations in the patients’ daily routine [7]. On the other hand, routine changes and removal from the familial home environment may cause distress and exacerbate behavior symptoms in dementia patients [60]. In this regard, telemedicine provides an efficient solution, allowing the assessment of the patient at home or the community clinic [78].

Telemedicine can significantly reduce traveled distance and time spent traveling compared to in-person visits [62,74,80]. During the COVID-19 pandemic, e-mail-based care for patients with dementia was proven feasible and effective [133]. The avoidance of unnecessary transportation, as well as the distance and time saved, benefit not only the patients but also their caregivers or family members that need to accompany them for the medical visits [83]. Importantly, telemedicine is beneficial for patients in advanced stages of dementia with mobility limitations, being bedridden or in a wheelchair, whose transportation is costly, stressful, laborious, and time-consuming [52,78]. Furthermore, telemedicine may reduce waiting time for appointments with specialists in the waiting room, resulting in higher convenience and less frustration [22].

Dementia care creates a significant economic burden for patients, families, caregivers, and healthcare systems [61]. Therefore, health policy planning and the development of cost-effective novel approaches are required. Telemedicine has also been shown to reduce the cost of medical visits [83]. The avoidance of unnecessary transportation and the reduction of travel time result in lower costs. In addition, there are free communication platforms available that can be utilized for telemedicine purposes after careful consideration of patients’ data safety [33]. The reduction of emergency department visits and unnecessary hospitalizations may also be associated with lower public healthcare costs [22]. Another study showed that videoconferencing was cost-effective for dementia diagnosis, in case the specialist should drive for more than two hours in order to deliver in-person service [134]. The FamTechCare intervention aims to provide dementia specialists feedback to caregivers based on video recordings and is cost-effective compared to telephone support interventions [135]. However, another study demonstrated that a remote caregiver support intervention only resulted in short-term cost savings, which could not be maintained for one year [136]. Therefore, in many cases, telemedicine use is associated with lower costs for the patients, family members, and the health system, as well as higher resource savings, allowing for more cost-effective and efficient care.

## 4. Discussion

Collectively, a growing body of evidence suggests that telemedicine may be a reliable and valuable tool for the care of patients with dementia. Regarding effectiveness, telemedicine can improve the accessibility to specialized care, especially for patients living in remote and underserved areas. Dementia specialists can reliably and effectively evaluate patients, neuropsychological testing can be provided, appropriate treatment recommendations can be suggested, and unnecessary emergency visits and hospitalizations may sometimes be prevented. Through telemedicine, patients can receive earlier diagnosis since travel and waiting time for the evaluation by a dementia specialist can be reduced. Telemedicine can also offer personalized care for patients’ and families’ needs and preferences, as well as cultural and ethnic/racial diversities, thereby contributing to patient-centeredness and equitability. The interconnection with community resources, the multidisciplinary team approach, the use of telerehabilitation services, and support and education for caregivers may also allow for improved integrated care. Throughtelemedicine, the infection risk is limited, which is a crucial safety issue, especially during the COVID-19 pandemic. Increased user convenience and reduced cost are some additional benefits in terms of efficiency.

However, it is essential to note that the number of visits, the stages of dementia of the participants, and the structure and elements of telemedicine visits across the established programs for dementia are highly variable [6]. For example, some programs provide video-based telemedicine care at home, while others at regional health centers or community clinics. Some programs include in-person visits for initial assessment, neuropsychological testing, or obtainment of vital signs, while others do not. Technical assistance is not available in all such telemedicine programs. In some cases, telemedicine is used only for initial evaluation, while in other cases, this model of care is applied for follow-up of diagnosed patients [7]. Moreover, a healthcare facilitator, a nurse, or a local physician, whose contribution is important for the remote assessment, was not always present during the telemedicine visits in the abovementioned studies. In addition, a control group (i.e., being evaluated in face-to-face visits) was not always used in the abovementioned studies. In many studies, the reasons for not participating in telemedicine visits have not been examined, potentially resulting in selection bias. Further, inter-rater variability is also an important factor to consider when comparing face-to-face and telemedicine visits in case the rater or physician is not the same in both situations. Therefore, comparisons between existing telemedicine models of care are hard to make, and conclusions regarding the reliability of the examination or improvement of care should be drawn with caution. The development of more appropriate methodological approaches to evaluate reliability, effectiveness, and efficiency is also needed.

Furthermore, most studies in telemedicine for dementia care are cross-sectional. Longitudinal studies are needed to investigate both the short-term and long-term effects of telemedicine on patients’ and caregivers’ outcomes compared to in-person visits [7]. Further, studies after the COVID-19 pandemic are also needed since the COVID-19 pandemic was characterized by specific conditions and challenges that may not be applicable to the period after the pandemic. Additionally, most studies have been conducted among well-educated individuals of high socioeconomic status. This limitation limits the generalizability of results to patients of lower socioeconomic backgrounds or educational levels [70]. Future research is needed in this direction, including participants from a wide range of socioeconomic statuses and levels of education. Furthermore, the low response rates regarding satisfaction among participants require caution since non-responding may be associated with lower satisfaction levels [70].

Although emerging literature evidence, including clinical trials, has demonstrated the value and promising potential of telemedicine services in dementia care, their implementation in daily clinical practice, dissemination, and effective incorporation into the health systems are still limited. For this purpose, alterations in healthcare policies are required. Funding opportunities and research grants for pilot activities in telemedicine are sometimes utilized, but the sustainability of these initiatives without continuous public support is often limited. Organizational, administrative, and technical challenges usually fall on the shoulders of primary care physicians, long-term care facilities, or caregivers without adequate support from the public health system [95]. Higher government investments and more active engagement by healthcare stakeholders, healthcare professionals, patients, and caregivers are required in this direction [33].

Regulatory issues and the absence of national legislation and reimbursement for telemedicine services in many countries is another important barrier that may prevent physicians from offering care remotely to their patients [151]. The lack of legal regulations regarding data privacy issues may also hinder the adoption of telemedicine by patients [52]. In the United States of America, licensure requirements may limit the provision of telemedicine dementia care across different states [7]. A study in Brazil indicated that physicians needed regulations to offer teleconsultations [152]. Online prescription, coverage, credentialing, medical malpractice, privacy, security, and fraud are some of the regulatory issues that need to be handled for the effective use of telemedicine services [22].

Currently, in most medical schools and residency neurology programs, physicians have no official training in telemedicine. The limited education of healthcare professionals in telemedicine serves as another obstacle to its broader application in the health systems. However, in this regard, the American Academy of Neurology has provided a published framework for developing a telemedicine educational curriculum for neurology residents [153].

Using a novel telemedicine care model may also initially receive significant resistance from patients, families, and caregivers, especially older individuals. Some studies have shown that patients perceive in-person care by primary care providers better than telemedicine visits [154]. Since primary care providers can largely influence their patients, they can discuss with them the benefits and restrictions of telemedicine, answer potential queries, discuss privacy concerns, and explain that telemedicine will not replace in-person care or limit their continuous contact with primary care providers, but rather provide an additional opportunity, thereby encouraging its future use [154]. Furthermore, blended approaches bringing together remote and in-person activities have been recommended as potential facilitators [33]. Another potential barrier to telemedicine is the fact that many individuals are unaware of the availability of telemedicine as an option, as well as the limited understanding among patients and family members about how to access telemedicine services [22,155]. Finally, ageism and stigma may also result in the de-prioritization of older individuals in telemedicine visits [156].

Access to the appropriate technological equipment, digital literacy, and the availability of an internet connection are also important issues to consider, especially in remote rural areas [157]. In the United States of America, it has been estimated that more than half of older individuals were not ready for video-based visits during the COVID-19 pandemic, mainly due to inexperience with technology [158]. The use of technical jargon for digital terms is also an obstacle, especially for the elderly [33]. The wide variability of the available telemedicine platforms may hinder the acceptability and eagerness of being trained to use them [33]. For older patients, in particular, it has been proposed that providing written detailed instructions on how to use telemedicine services may help in this direction [158]. Training older individuals in digital tools may also be beneficial [33], and the use of understandable terms is also very important [33]. Caregivers also suggest that one technological barrier is that older patients with dementia have limited ability to manage the equipment and engage in remote programs without assistance [159]. Cognitive impairment is associated with loweruse of technology in older individuals [160].

Low income and educational levels have been associated with inadequate access to digital technologies; hence, technical assistance should be provided, especially in these cases [161]. Furthermore, the younger caregivers’ age has been associated with higher rates of the feasibility of telemedicine visits for patients with dementia [162], suggesting that the experience of the caregivers with technology plays an important role. Some potential solutions include providing technological equipment, such as tablets or laptops, as well as ensuring free internet access to all [33].

Older patients often have significant concerns regarding privacy and confidentiality, and the use of secure software is very important. It should be clearly explained to patients to show their medical data are securely transferred and shared, as well as who has access to them [33]. Until now, older individuals’ views and perspectives have not been adequately included in the design of telemedicine interventions [163]. However, the development of age-friendly telemedicine services adapted to the needs of older patients is of paramount importance [33]. Patients’ preferences, prior experiences, and perspectives that may affect technology acceptability have not been studied among individuals with cognitive impairment yet [70]. Future studies are needed in this direction since these factors may significantly affect the feasibility and acceptance of telemedicine services. In this context, a personalized assessment of the users’ needs before the telemedicine visits could allow for adaptations regarding equipment, hearing, or visual impairment [70].

Apart from the patients, physicians may be skeptical about the use of telemedicine, and there is evidence showing that they may be less satisfied with telemedicine compared to patients [7]. Some healthcare providers are also not adequately familiarized with telemedicine platforms [33]. Primary care physicians are satisfied with telemedicine services for dementia [164,165]. However, there is also evidence showing that healthcare professionals may not recognize the benefits of telemedicine for older patients with complex conditions, including cognitive impairment, based on the assumption that these patients may not be able to understand the instructions and effectively participate in the remote visits [166,167]. Discussions with healthcare professionals about their concerns and the exploration of potential solutions to mitigate these concerns have been recommended [33]. The adherence of general practitioners to teleconsultations’ recommendations for older individuals in nursing homes has been associated with depressive symptoms [168], suggesting that the psychological effects of telemedicine on general practitioners should also be considered and further investigated.

Interrupted or delayed internet connection may also create difficulties. It can also interfere with neurological examination, especially regarding the evaluation of movement disorders, such as bradykinesia and tremor, which are sometimes important for the differential diagnosis of dementia subtypes. The availability of an IT technician, the adoption of a timely and simple backup process in case of connectivity failure, and the use of the same platform for all remote consultations are practical solutions [33,169].

Hearing loss in older ages is prevalent in up to 90% of patients with dementia and often remains untreated [13]. Hearing difficulties may challenge cognitive assessment [13]. Visual impairment is also very frequent among patients with dementia, and it is estimated to affect up to 30% of this population [13]. Hearing and visual impairment restrict the use of telemedicine since it may hinder effective physician–patient communication, and the patients may have difficulties hearing or seeing the instructions or the stimuli of the neuropsychological tests via video [33,170]. However, attempts for technological adaptations to visual or hearing impairment in the telemedicine environment are limited [13].

Although the validity of several neuropsychological tests has already been investigated in telemedicine settings, more work is needed for specific tests for various dementia forms and different disease stages.

Apart from the care of patients with cognitive impairment, telemedicine may also facilitate prevention strategies, raising awareness about AD and other forms of dementia, cognitive screening, and the increased participation of patients in clinical trials, especially for screening and recruitment [52]. This is especially important for patients in remote areas with limited opportunities to engage in clinical trials, which are usually conducted in university hospitals and big cities.

“Even though this was not a systematic review, our aim was to provide, for the first time, an initial map of the potential role of telemedicine in the improvement of quality of healthcare for patients with dementia, based on the WHO dimensions. For this purpose, we critically discuss available literature evidence, highlighting gaps and potential challenges for future research in this field”.

## 5. Conclusions

In summary, telemedicine is a reliable and valuable tool for the care of individuals with cognitive impairment in AD or related forms of dementia. It has the capacity to improve effectiveness, timeliness, patient-centeredness, integrated care, efficiency, and equitability. It gives the opportunity for increased access to specialized healthcare, especially for patients living in underserved remote areas. This opportunity allows for earlier diagnosis, appropriate treatment, and fewer visits to the emergency departments and hospitalizations. Moreover, telemedicine allows for a multidisciplinary treatment approach and can improve personalized care by focusing on patients’ and families’ needs, preferences, and cultural, ethnic, and racial diversities. It is associated with high satisfaction rates and increased convenience for users. It can also provide support for caregivers, connection with community resources, education of the patients, caregivers, and primary care physicians, as well as increased access of patients to clinical trials. Furthermore, telemedicine may result in reduced cost and unnecessary transportation and lower infection risk.

However, significant challenges include legislative and regulatory aspects, resistance from patients, caregivers, and physicians, ageism and stigma, limited education of physicians in telemedicine, digital illiteracy, technological equipment or internet connection issues, hearing or visual impairment, limited awareness regarding the availability of telemedicine services, and lack of sustained support from the public sector.

In this context, further attempts are needed to investigate and overcome relative barriers to the implementation of telemedicine in daily clinical practice. Nevertheless, telemedicine provides a very useful way to address the emerging need for better quality of care for patients with dementia worldwide, and the public sector should invest more resources in its successful integration into the health systems.

## Figures and Tables

**Figure 1 medicina-58-01705-f001:**
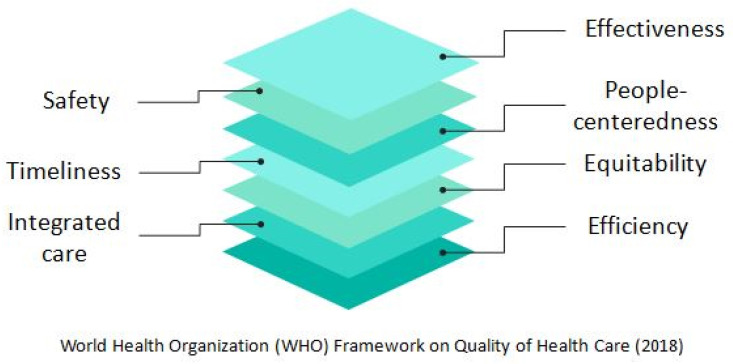
The seven dimensions of Quality of Health Care, according to World Health Organization Framework, 2018.

**Table 1 medicina-58-01705-t001:** Healthcare quality dimensions according to the World Health Organization (WHO), 2018, its definition, and examples of the value of each dimension of quality of care in Alzheimer’s disease and related forms of dementia.

Health Care Quality Dimension	Definition	Examples of the Importance of High Quality of Care in Alzheimer’s Disease and Related Dementias	References
Effectiveness	The provision of evidence-based health services to individuals who need them	Lack of education and training of primary care physicians in the diagnosis and treatment of dementia	[4,34]
		The differential diagnosis of dementia causes has important clinical impact	[35]
		The differential diagnosis between dementia mimics has important clinical impact	[36]
		Identification of atypical clinical presentations and rarer forms of dementia, especially at younger agesis challenging	[36]
		Detailed neuropsychological testing is often required for the accurate dementia diagnosis, especially at early stages	[37]
		The limited time of primary care physicians does not allow sufficient discussion, counseling, and personalized management plan	[34]
		Adherence to dementia guidelines and evidence-based recommendations is associated with better overall quality of care	[38]
		Adherence of primary care physicians to dementia guidelines and quality care indicators is inadequate	[38,39,40]
		Most patients with dementia have inadequate access to appropriate formal care services	[41]
		The participation of physicians in educational seminars on dementia care is associated with improved quality of life	[38]
Safety	The avoidance of harmto people for whom the care is intended.	Traveling to specialized physicians is challenging for dementia patients due to their cognitive and mobility issues	[7]
		COVID-19 poses high infection risks in traditional in-person clinical settings in the elderly	[33]
		During the COVID-19 pandemic, many older patients with dementia do not receive appropriate medical care	[42]
		Inappropriate and high prescription of antipsychotic drugs for patients with dementia is high	[43,44]
		Improper use of antipsychotics is associated with a higher risk of death and ischemic events in the elderly	[45]
		AD diagnosis is associated with higher COVID-19-associated mortality	[46]
People-centeredness	The delivery of care that respects and responds to personal needs, values, and preferences	Individualized interventions by primary care physicians in collaboration with dementia care managers are associated with lower caregiver stress and behavioral symptoms	[6,47,48]
		Personalized evaluation and management are associated with higher-quality dementia care	[34]
		Care models with shared decision making is associated with improved satisfaction	[49]
		Patients’ and family members’ preferences, perceptions, and needs are an integral part of dementia care	[49]
Timeliness	The reduction of waiting times and delays that could be harmful not only for those who receive but also give care	Diagnostic sensitivity of dementia is correlated with the frequency of contact between patients and providers	[4]
		Regular monitoring contributes to a better quality of life	[38,43]
		Specialized neurologists and memory clinics are lacking in remote areas	[50]
		Earlier detection of cognitive decline is associated with better health outcomes	[49]
		AD prevalence is higher in rural areas	[51]
Equitability	The provision of care that is independent of age, race, ethnicity, geographical location, socioeconomic status, sex, gender, religion, political or linguistic affiliation	Limited recruitment of physicians at rural health centers	[22]
		Lack of transportation infrastructure and socioeconomic disparities limit the accessibility of patients living in rural areas	[52]
		Often long travel distances for appropriate access to specialized care	[53]
		Compared to non-Hispanic Whites, a greater percentage of non-Hispanic Blacks and Hispanics had a missed or delayed dementia diagnosis	[54]
Integrated care	Coordinated care across different levels and providers, allowing the availability of all appropriate health services throughout life course	Patients, caregivers, and family members often require referral for legal issues, advice on long-term facilities, improvement of home environment, psychological support, information about available services, support groups, educational resources, and administrative assistance	[34]
		Cooperation between different healthcare professionals for dementia care is limited, and community-based organizations are currently underutilized and inadequately incorporated into the healthcare system	[34,43]
		Social worker engagement improves the quality of care for dementia	[38]
		Transdisciplinary collaborative team care (physicians, neuropsychologists, social workers, registered nurses, and nurse practitioner managers) and linkages to appropriate community resources are associated with better quality of care, improved counseling, reduced caregivers’ stress and patients’ behavioral and depressive symptoms, and fewer hospitalizations and visits to the emergency departments	[38,55,56]
		In integrated care, team members contribute with their expertise and clinical or management strengths for appropriate dementia care	[49]
		Fragmented care and difficult-to-navigate healthcare services are significant barriers to treatment	[57]
		The integration of community-based organizations (e.g., Alzheimer’s Association) into the health systems improves quality of care for patients with dementia	[55]
		Participation of nurse practitioners in care is associated with higher healthcare quality for dementia patients, reduced risk of falls and incontinence, and better adherence to care recommendations	[58,59]
Efficiency	Maximizing the benefit of available care resources and avoiding relevant waste	Fewer emergency department visits and unnecessary hospitalizations may be associated with lower public healthcare costs	[22]
		Changes in routine and home environment may cause anxiety and exacerbate behavior symptoms in dementia patients	[60]
		Dementia care creates significant economic burden for patients, families, caregivers, and healthcare systems	[61]

**Table 2 medicina-58-01705-t002:** Healthcare quality dimensions according to the World Health Organization (WHO), 2018, and the role of telemedicine in each of them dementia care.

Health Care Quality Dimension	The Role of Telemedicine in Each of the Quality of Care Dimensions in Alzheimer’s Disease and Other Related Dementias	References
Effectiveness	Telemedicine is effective in confirming or providing a diagnosis for cognitive impairment	[27,62]
	Alterations in drug prescriptions were recommended via telemedicine in more than 1/3 of patients (longitudianl study, 3-year follow-up period, 45 clinical video telehealth encounters)	[62]
	Rural community clinics can be effectively connected through teleconsultations with physicians specialized in dementia care in University Hospitals (longitudinal study, 188 patients with dementia, face-to-face versus telemedicine care)	[63]
	Telephone-based remote care is feasible for younger patients with dementia (retrospective study for a 2-year period, 1121 calls)	[64]
	Telemedicine provided by specialists is associated with 1.8 and 1.1 medication alterations for patients with dementia at initial assessments and follow-up visits, respectively, for 12-month period (longitudinal study, 199 clinical video telehealth patient encounters)	[65]
	Teleconsultation in dementia is associated with treatment modifications at approximately 10%, especially for those with AD or living with a relative (multicenter study, 874 patients)	[66]
	Telemedicine use is associated with longer treatment duration and compliance in dementia patients during a 5-year period (259 patients in-person, 168 patients via telemedicine)	[67]
	Telemedicine is feasible for follow-up and ongoing care	[50,68]
	Fewer canceled medical visits and improved transitions between the follow-up clinic and primary care supported by a case manager or geriatric assessor via telemedicine (55 telemedicine sessions)	[68]
	Telemedicine via video-conferencing are associated with improved quality of life, better physical and mental health, less perceived burden, and higher self-efficacy, compared to only telephone-based visits among patients with neurodegenerative diseases	[42,69]
	Telemedicine via video-conferencing may improve the well-being and resilience of patients (self-efficacy, perceived burden) with neurocognitive disorders and caregivers and avert MoCA deterioration (60 older adults with neurocognitive disorder; supplementary telehealth via video conference vs. via telephone)	[60]
	Telemedicine is a valid triage tool for patients with frontotemporal dementia regarding clinical worsening (CDR-FTD scale), change in quality of life, and COVID-19 symptoms, with high satisfaction of the caregivers(26 telemedicine clinical interviews with caregivers, 4 with both patients and caregivers	[70]
	Telemedicine for acute illnesses is associated with less unnecessary visits to emergency departments among older adults with dementia in senior living communities (1 year of access to telemedicine is associated with a 24% reduction in emergency department visits)	[71]
	Emergency department use was reduced for ambulatory-care sensitive conditions after the introduction of telemedicine for older individuals in senior living communities (prospective cohort study at a primary care geriatrics practice)	[72]
	Videoconferenced geriatric medicine grand rounds on a weekly basis are feasible and beneficial for healthcare professionals in 9 urban and 14 remote rural areas (questionnaire: reason of attendance, evaluations of presentations)	[73]
Safety	Specialists could identify inappropriate drug use that might contribute to cognitive decline in almost half of the visits through telemedicine (interprofessional dementia assessment by a geriatrician, geropsychologist, geriatric psychiatrist or neurologist, and social worker using clinical videotelehealth technology)	[74]
	Telemedicine through video-conferencing is associated with better quality of life for patients with dementia compared to only telephone-based visits during the social isolation of the COVID-19 pandemic (60 older adults with neurocognitive disorder; supplementary telehealth via video-conference vs. via telephone)	[60]
	In case of home-based video-conferencing, telemedicine allows the physician to directly observe the home environment of the patient and suggest alterations such as individualized recommendations for the prevention of falls (feasibility study, 10 videoconferencing visits)	[75]
People-centeredness	Patients and caregivers accept telemedicine as a very convenient model of care. Patients with AD and their caregivers are very satisfied with telemedicine (overall satisfaction rates 88–98%)	[62,76,77,78]
	Telemedicine is preferred over in-person visits	[76,79]
	Similar satisfaction rates are observed between telemedicine and traditional in-person visits (230 participants recruited from outpatient dementia clinic)	[80]
	Telemedicine allows for the identification of caregivers’ needs (rural caregiving telemedicine program, 1-year questionnaire on risk factors, behavioral management, diagnosis, and medications)	[81]
	Semi-structured interviews for the experiences of patients and caregivers on telemedicine demonstrated that although proactive teleconsultations during the COVID-19 pandemic are effective, they should be focused on needs and practical recommendations (community-based patients living with dementia (30) and their carers (31))	[82]
	Telemedicine allows family members living away from the patient’s home to attend the video-conference visit, allowing for shared decision making (older participants, 72.1% with cognitive impairment, 32 patient evaluations, 80 clinician feedback evaluations, satisfaction, care access during pandemic, and travel and time savings)	[83]
	Assistive technology use and telecare for individuals with dementia are not associated with prolonged time of independent living (randomized controlled trial, 495 participants)	[84]
Timeliness	Telemedicine allows for real-time medical reporting and sharing, thereby avoiding unnecessary delays (videoconferencing 28 patients from outpatient clinic)	[24]
	The use of wearable devices, remote monitoring sensors, or web-based platforms may facilitate early detection of medical emergencies and timely intervention	[33,85]
	Telemedicine reduces waiting times for appointments with specialized physicians, allowing earlier diagnosis and treatment of dementia-related various medical complaints(60 older adults with neurocognitive disorder; supplementary telehealth via video conference vs. via telephone)	[60]
Equitability	Telemedicine in rural areas is effective, with high satisfaction rates, allowing for better access to timely care, reduced cost, and avoidance of unnecessary transportation	[22,86,87]
	Telemedicine facilitates the elimination of geographical disparities, allowing patients with dementia from rural and urban areas to access specialized healthcare	[53,88]
	Digital literacy, lower education level, and worse cognitive function are associated with less engagement in remote interventions promoting lifestyle modifications among older adults	[89,90]
Integrated care	In a Tennessee-based program, specialists recommended referrals to social workers and the use of long-term care services in almost two-thirds of the telemedicine visits	[62,76]
	A Pittsburgh-based telemedicine program for dementia care, including a geriatrician, geriatric psychiatrist, psychologist, social worker, and nurse manager, is highly acceptable and successful fur rural areas (patient satisfaction survey, 156 clinic visits)	[74]
	Telemedicine may significantly enable interdisciplinary dementia care	[86,87,91,92,93]
	The use of a nurse practitioner-led mobile memory clinic incorporated in the general practice targeting patients with poor socioeconomic status and limited access to care is feasible and acceptable (1-year, 102 patients)	[94]
	Telemedicine can aid in the assessment and management of psychotic symptoms of patients with neurodegenerative disorders in long-term care facilities (multidisciplinary consensus panelist of best practices in telemedicine for patients with dementia-related psychosis or Parkinson’s disease-related psychosis)	[95]
	Telemedicine allows telerehabilitation for patients with AD dementia, frontotemporal dementia, and mild cognitive impairment	[96,97,98,99]
	Computerized cognitive training among patients with or at risk for dementia is effective	[100,101,102]
	Speech therapy is effective in primary progressive aphasia and alexia	[103,104]
	Virtual reality for patients with dementia is associated with reduced neuropsychiatric symptoms (i.e., depression and agitation, apathy) and quality of life	[105,106,107]
	Tele-exercise programs through video conferencing are feasible and acceptable among patients with AD and their caregivers	[108,109]
	Video-based caregiver support for stress, education, and training for behavioral symptoms are feasible and effective	[110,111,112,113,114,115,116,117,118,119,120,121,122,123,124,125,126,127]
	Remote telephone-based cognitive behavioral therapy to caregivers of patients with AD for the enhancement of physical or mental health is effective (273 family caregivers, 50-min sessions)	[128]
	Remote cognitive behavioral therapy to caregivers of patients with AD for insomnia treatment is also effective (four-session CBT-I protocol)	[129]
	Most caregivers are satisfied with the FamTechCare service, which allows for tailored expert feedback based on video recordings (multisite randomized controlled trial, satisfaction survey)	[130]
	Subjective burden levels of the caregivers have not been significantly affected by a telehealth-based intervention, while objective measures of activity and sleep showed a slight decline	[131]
	Assistive technology and telecare are not associated with reduced caregivers’ burden (randomized-controlled trial)	[132]
Efficiency	Telemedicine reduces traveled distance and time spent traveling compared to in-person visits	[62,74,80]
	Telemedicine is beneficial for patients in advanced stages of dementia with mobility limitations, being bedridden or in a wheelchair, whose transportation is costly and time-consuming	[52,78]
	The avoidance of unnecessary transportations and the distance and time saved have significant effects on patients, caregivers, and family members that need to accompany them for medical visits(older participants, 72.1% with cognitive impairment, 32 patient evaluations, 80 clinician feedback evaluations, satisfaction, care access during pandemic, and travel and time savings)	[83]
	During the COVID-19 pandemic, e-mail-based care for patients with dementia is feasible and effective (retrospective analysis, 14-month period, 374 e-mails sent by 213 patients)	[133]
	Videoconferencing is cost-effective for dementia diagnosis, in case the specialist should drive for more than two hours in order to deliver in-person service (break-even analysis)	[134]
	The FamTechCare intervention aiming to provide dementia specialists feedback to caregivers based on video recordings is cost-effective, compared to telephone support interventions (clinical trial, cost-effectiveness analysis)	[135]
	A remote caregiver support intervention only resulted in short-term cost savings, which could not be maintained for one year (randomized controlled trial)	[136]

## Data Availability

Not applicable.

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
