# Peer review of "How Telemedicine Can Improve the Quality of Care for Patients with Alzheimer’s Disease and Related Dementias? A Narrative Review"

_medicina, 2022, doi:10.3390/medicina58121705_

Round 1
Reviewer 1 Report
This is a large scale and comprehensive systematic review article.
It discussed how telemedicine can improve the quality of healthcare for AD and related dementias regarding the seven dimensions of healthcare quality defined by the World Health Organization (WHO). Readers can use this article as a map to every piece of evidence on each aspects of healthcare quality including effectiveness, safety, people-centeredness, timeliness, equitability, integrated care, and efficiency. However, There is lacking of clearly explain the technical aspects , which is partly because telemedicine is rather a collective noun than a specific procedure.
Author Response
Answers to Reviewers Comments
Reviewer 1
This is a large scale and comprehensive systematic review article.
It discussed how telemedicine can improve the quality of healthcare for AD and related dementias regarding the seven dimensions of healthcare quality defined by the World Health Organization (WHO). Readers can use this article as a map to every piece of evidence on each aspects of healthcare quality including effectiveness, safety, people-centeredness, timeliness, equitability, integrated care, and efficiency. However, There is lacking of clearly explain the technical aspects, which is partly because telemedicine is rather a collective noun than a specific procedure.
Answer:
We would like to thank very much the Reviewer for the time and effort to read our manuscript, and the positive feedback on our work. Regarding the important comment about the description of the technical aspects of telemedicine, as the Reviewer also mentioned, telemedicine is indeed a general term that incorporates a wide range of methods through which healthcare can be remotely provided to patients by healthcare professionals(Mechanic OJ, Persaud Y, Kimball AB. Telehealth Systems. 2022 Sep 12. In: StatPearls. Treasure Island (FL): StatPearls Publishing; 2022 Jan–. PMID: 29083614). Based on this definition, telemedicine is an “umbrella” term and not a specific procedure.
Since the aim of this review was not to describe in detail the different technical aspects of telemedicine, we had previously briefly mentioned the different telemedicine types in the last two paragraphs of the Section “3.2. Definition and primary forms of telemedicine”. However, we agree with the Reviewer that a more detailed description of the diverse technical types of telemedicine would be useful. For this purpose, we made some modifications at the 3.2 Section.
In addition, we would like to mention that based on the other two Reviewers’ comments, we altered the title of our review to “How telemedicine can improve the quality of care for patients with Alzheimer’s disease and related dementias? A systematized narrative review” in order to be more clear about the methodology we followed. Even though we searched the literature in a systematic manner, our aim was not to conduct a systematic review. Given the vague definition of “systematized review” (Grant MJ, Booth A. A typology of reviews: an analysis of 14 review types and associated methodologies. Health Info Libr J. 2009 Jun;26(2):91-108. doi: 10.1111/j.1471-1842.2009.00848.x. PMID: 19490148.) , we chose to change the type of our article to “narrative”, in order not to create confusion. We believe that the Reviewer understands the reasons for this specific alteration in the Title, and we are willing to provide further explanations if needed.

Reviewer 2 Report
I suggest the author present flowchart diagram of searching method based on PRISMA Statement. Second, please give assessment of the risk of bias from the reviewed articles to assess quality of the articles. Third, the author can register this review to Prospero (www.crd.york.ac.uk).
Author Response
Reviewer 2
I suggest the author present flowchart diagram of searching method based on PRISMA Statement. Second, please give assessment of the risk of bias from the reviewed articles to assess quality of the articles. Third, the author can register this review to Prospero (www.crd.york.ac.uk).
Answer:
We would like to greatly thank the Reviewer for the important comments. However, we would like to note that our aim was not to conduct a systematic review, even though our research was done in a systematic manner. According to the definition of “systematized reviews”, they attempt to include one or more elements of the systematic review process while stopping short of claiming that the resultant output is a systematic review. They may identify themselves parenthetically as a ‘systematic’ review. They may or may not include comprehensive searching, and they may or may not include quality assessment. (Grant MJ, Booth A. A typology of reviews: an analysis of 14 review types and associated methodologies. Health Info Libr J. 2009 Jun;26(2):91-108. doi: 10.1111/j.1471-1842.2009.00848.x. PMID: 19490148.). We have not used the strict PRISMA guidelines, and not made a quality assessment of the studies. In addition, since our study was not a systematic review, we cannot register it in PROSPERO too.
However, we understand that the definition of “systematized review” is a bit vague and creates confusion. Therefore, in order to be more clear about the methodology we used, we chose to alter the title of our manuscript as follows: “How telemedicine can improve the quality of care for patients with Alzheimer’s disease and related dementias? A systematized narrative review”. We hope the Reviewer understands the reasons for this specific alteration in the Title, and we are willing to provide further explanations if needed.

Reviewer 3 Report
Telemedicine or Onlinemedicine becomes popular, especially during the COVID-19 pandemic. In this paper, the author gives a critical review of “How telemedicine can improve the quality of care for patients with Alzheimer’s disease and related dementias?” Obviously, it is an interesting and important topic, which is need to be further discussed. In this review, the author aims to analyze and discuss how telemedicine can improve the quality of healthcare for AD and related dementias in a structured manner, based on the seven dimensions of healthcare quality defined by the World Health Organization. I have some comments listed below:
First of all, the content of Tables 1 and 2 are already described in the main paragraph of the result and discussion section, for example, In the table1 it is shown that "Many primary care physicians are not familiarized with the diagnostic criteria of dementia, and they lack appropriate education and expertise in assessing and treating patients with cognitive impairment", the exact same sentence also shown in line 238-240. You could not just copy all sentences in the main paragraph to the table. The content of the tables needs to be the details of the paper you cited, which tell us more about what part is important there. For example, where the research is done, how many people are participating, and how long it going. What special method did they use to make the research work?
Secondly, from the result, I don’t see any statistical analysis for the materials themselves which is important for this review. For example, how many papers you finally collect, when they were published, and what type of publications. the paper is clinical research or a common survey or a review? You need to provide this information to us to help us know the bases.
Third, In 3.1. Defining the concept and dimensions of healthcare quality part, it is confusing which definition or criteria you want to use here. The IMOs or WHO, look to describe the same things.
Last, in the paper, the author uses a lot of long sentences in this review, which makes it difficult to read and understand. for example, in the abstract, they use too many “and” in one sentence, which makes me confused.
All in all, I suggest the author rewrite the whole paper and do a more deep analysis of the data you collected and resubmit the paper.Author Response
Reviewer 3
Telemedicine or Onlinemedicine becomes popular, especially during the COVID-19 pandemic. In this paper, the author gives a critical review of “How telemedicine can improve the quality of care for patients with Alzheimer’s disease and related dementias?” Obviously, it is an interesting and important topic, which is need to be further discussed. In this review, the author aims to analyze and discuss how telemedicine can improve the quality of healthcare for AD and related dementias in a structured manner, based on the seven dimensions of healthcare quality defined by the World Health Organization. I have some comments listed below:
Answer:
We greatly appreciate the Reviewer’s time and effort carefully reading our manuscript, as well as the important points raised, which we address below, and will significantly improve this work.
First of all, the content of Tables 1 and 2 are already described in the main paragraph of the result and discussion section, for example, In the table1 it is shown that "Many primary care physicians are not familiarized with the diagnostic criteria of dementia, and they lack appropriate education and expertise in assessing and treating patients with cognitive impairment", the exact same sentence also shown in line 238-240. You could not just copy all sentences in the main paragraph to the table. The content of the tables needs to be the details of the paper you cited, which tell us more about what part is important there. For example, where the research is done, how many people are participating, and how long it going. What special method did they use to make the research work?
Answer:
We would greatly appreciate this important point raised by the Reviewer. We made several alterations in the Table 1 and Table 2, in order not to provide exactly the same information in the text and tables. We also incorporated some additional details for each study especially in Table 2 (which is the main focus of our review), regarding the study design, number of participants, number of telemedicine visits, and follow-up period (when applicable).
Secondly, from the result, I don’t see any statistical analysis for the materials themselves which is important for this review. For example, how many papers you finally collect, when they were published, and what type of publications. the paper is clinical research or a common survey or a review? You need to provide this information to us to help us know the bases.
We thank the Reviewer for this comment. However, we would like to note that our aim was not to conduct a systematic review, meta-analysis or any statistics, even though our research was done in a systematic manner. Based on the definition of “systematized review”, this kind of reviews attempt to include one or more elements of the systematic review process while stopping short of claiming that the resultant output is a systematic review. They may identify themselves parenthetically as a ‘systematic’ review. They may or may not include comprehensive searching, and they may or may not include quality assessment. (Grant MJ, Booth A. A typology of reviews: an analysis of 14 review types and associated methodologies. Health Info Libr J. 2009 Jun;26(2):91-108. doi: 10.1111/j.1471-1842.2009.00848.x. PMID: 19490148.). We have not used the strict PRISMA guidelines, so, we did not mention that our review was systematic, but systematized. In addition, we have not conducted detailed data extraction to provide the information asked by the Reviewer, which would be done if our review was systematic.
However, we understand that the definition of “systematized review” is a bit vague and creates confusion. Therefore, in order to be more clear about the methodology we used, we chose to alter the title of our manuscript as follows: “How telemedicine can improve the quality of care for patients with Alzheimer’s disease and related dementias? A systematized narrative review”. We hope the Reviewer understands the reasons for this specific alteration in the Title, and we are willing to provide further explanations if needed.
Third, In 3.1. Defining the concept and dimensions of healthcare quality part, it is confusing which definition or criteria you want to use here. The IMOs or WHO, look to describe the same things.
Answer: We appreciate the Reviewer for this comment. Indeed, WHO definition of healthcare quality is based on IMO definition, and that is the reason we firstly mentioned IMO definition. We have already mentioned that our review is based on WHO definition. However, in order to be more clear, we made the following alterations in the third paragraph of “3.1. Defining the concept and dimensions of healthcare quality”
“Similarly, as recently stated in the World Health Organization (WHO) Framework on Integrated People-centered Health Services–which was based on IOM framework-, there is an emerging acknowledgment that “high-quality care” involves “care that is safe, effective, people-centered, timely, efficient, equitable and integrated” (Figure 1).
Last, in the paper, the author uses a lot of long sentences in this review, which makes it difficult to read and understand. for example, in the abstract, they use too many “and” in one sentence, which makes me confused.
Answer: We thank the Reviewer for this note. We made several alterations in the Abstract and main text, in order to shorten the sentences, remove “and” when unnecessary and make them more clear and easily read.
All in all, I suggest the author rewrite the whole paper and do a more deep analysis of the data you collected and resubmit the paper.
Answer: We made several alterations in the main text, tables, and also provided a deeper analysis in some parts in the Discussion Section, based on the Reviewer’s comments.
For example, we made the following alterations:
2nd Paragraph of the Discussion: “In addition, a control group (i.e. being evaluated in face-to-face visits) was not always used in the abovementioned studies.”
2nd Paragraph of the Discussion: “In many studies, the reasons for not participating in telemedicine visits have not been examined, potentially resulting in selection bias.”
2nd Paragraph of the Discussion: “Further, inter-rater variability is also an important factor to consider when comparing face-to-face and telemedicine visits in case the rater or physician was not the same in both situations.”
2nd Paragraph of the Discussion: “The development of more appropriate methodological approaches to evaluate the re-liability, effectiveness and efficiency are also needed.”
3rd Paragraph of the Discussion” “Further, studies after COVID-19 pandemic are also needed, since COVID-19 pandemic was characterized by specific conditions and challenges that may be not applicable to the period after the pandemic.”
We added this comment at the end of the Discussion in our manuscript, to highlight the strength of this critical review:
“Even though this was not a systematic review, our aim was to provide, for the first time, an initial map of the potential role of telemedicine in the improvement of quality of healthcare for patients with dementia, based on the WHO dimensions. For this purpose, we critically discuss available literature evidence, highlighting gaps and potential challenges for future research in this field.”
Round 2
Reviewer 3 Report
The author put a lot of effort into revising table 1 and table 2, now it looks better. I suggest that you could check if it still has any grammatical mistakes.